# A Novel Active Contours Model for Environmental Change Detection from Multitemporal Synthetic Aperture Radar Images

**Salman Ahmadi [1],\* and Saeid Homayouni [2]** 

[1]  Department of Civil Engineering, Faculty of Engineering, University of Kurdistan, Sanandaj 6617715175, Iran
[2]  Centre Eau Terre Environnement, Institute National de la Recherche Scientifique,
    Quebec, QC G1K 9A9, Canada; saeid.homayouni@ete.inrs.ca
\*  Correspondence: s.ahmadi@uok.ac.ir

**Abstract:** In this paper, we propose a novel approach based on the active contours model for change detection from synthetic aperture radar (SAR) images. In order to increase the accuracy of the proposed approach, a new operator was introduced to generate a difference image from the before and after change images. Then, a new model of active contours was developed for accurately detecting changed regions from the difference image. The proposed model extracts the changed areas as a target feature from the difference image based on training data from changed and unchanged regions. In this research, we used the Otsu histogram thresholding method to produce the training data automatically. In addition, the training data were updated in the process of minimizing the energy function of the model. To evaluate the accuracy of the model, we applied the proposed method to three benchmark SAR data sets. The proposed model obtains 84.65%, 87.07%, and 96.26% of the Kappa coefficient for Yellow River Estuary, Bern, and Ottawa sample data sets, respectively. These results demonstrated the effectiveness of the proposed approach compared to other methods. Another advantage of the proposed model is its high speed in comparison to the conventional methods.

**Keywords:** SAR images; multitemporal; environmental change detection; active contours model

## 1. Introduction

Change detection is the process in which two remote sensing images of a region at different times are used to extract areas that have been changed during the time between the images. In the last decades, optical and radar remote sensing imageries have become vital resources in change detection applications due to their high spatial and temporal resolutions, useful spectral or polarimetric characteristics, extensive spatial coverage, and cost-effectiveness. This includes the synthetic aperture radar (SAR) images, owing to their ability in imaging in all weather conditions, such as rainy and dusty air and in the night, and also their suitable spatial resolution. They have been widely exploited in change detection applications for natural hazards' impacts [1–3], monitoring and mapping of environment and natural resources [4,5], urban development [6–8], etc.

The change detection algorithms can be categorized into supervised, semi-supervised, and unsupervised classes, depending on the availability of ground truth information. Supervised approaches, however, do have higher accuracy; they need sample data of change and unchanged areas to train the classifier model [9–11]. These models are less frequently used due to a lack of prior information in real applications. Additionally, the labeled and unlabeled samples are utilized simultaneously in the semi-supervised change detection algorithms [12,13]. In these algorithms, the labeled sample data are training data, and the rest of the pixels are the unlabeled sample data. In general, supervised and

semi-supervised algorithms have a better efficiency than unsupervised methods. However, due to the unnecessary usage of training data, the unsupervised approaches are more prevalent in change detection applications [14].

The unsupervised change detection methods can be summarized into two classes: Threshold-based and classification methods. In threshold-based algorithms, the goal is to find a threshold in order to classify the difference image into changed and unchanged classes correctly [15,16]. Gong et al. introduced a neighborhood-based ratio operator to generate a difference image and finally extracted the changed regions by applying a threshold on the difference image [17]. Furthermore, Sumaiya and Kumari calculated a threshold based on the mean of the difference image and the logarithm of two input SAR images to extract the changed areas from the background [18]. In another study, they proposed a change detection algorithm based on the Gabor filter and Kittler–Illingworth thresholding algorithm [19]. Although the threshold-based methods have higher simplicity and speed, they are less accurate than other methods.

According to the suitable accuracy and simplicity of the clustering classification-based models, they have been widely used in change detection from temporal SAR images. Classification-based methods are usually combined with conventional image processing and optimization algorithms to increase the accuracy, which reduces their computational cost [20,21]. Shang et al. utilized an artificial immune multi-objective clustering algorithm based on the intensity and texture of the differences image to identify changed regions from unchanged ones [22]. Additionally, Zheng et al. used a K-means clustering method to detect changed pixels using a linear combination of subtraction and log-ratio difference images [23]. Li et al. proposed a multi-objective fuzzy clustering algorithm in which the change detection problem is modeled as a multi-objective optimization problem with two objective functions to preserve image details and remove the noise [24]. Zheng et al. used an unsupervised saliency-guided method and K-means clustering to detect the changed areas in a difference image obtained from a logarithmic ratio operator [14]. In addition, Tian et al. developed an edge-based fuzzy clustering algorithm to detect the changes in the SAR images [25].

The artificial neural networks as a classification method are utilized in change detection applications. Although artificial neural networks can obtain a better performance, they have high complexity and low speed. Convolutional neural networks (CNNs), sparse auto-encoder, and unsupervised clustering algorithms were applied for detecting changes in SAR images [26]. Additionally, Li et al. used a deep learning network, named PCANet (Principal Component Analysis Network), and saliency method for SAR change detection [27]. Recently, Chen et al. introduced a fast unsupervised deep neural network to generate a difference image for change detection of SAR images [28].

Furthermore, other classification models, such as graph cuts, have been used as a change detector of temporal SAR images. Carincotte et al. developed a fuzzy hidden Markov chain model by combining two fuzzy and statistical points of view to identify changes in the SAR images [29]. Ma et al. fused wavelet coefficients for low- and high-frequency bands using fusion rules based on weight averaging and the minimum standard deviation, respectively, to extract changed regions from temporal SAR images [30]. Gong et al. presented a change detection method based on texture and intensity information and a multivariate generalized Gaussian graph cuts model [31].

In this paper, we developed an innovative model based on the active contour model for change detection from the SAR images. The proposed model consists of three stages: The difference image generation, change detection by active contours, and accuracy assessment of the model. Firstly, a new difference image operator is introduced to increase the accuracy of change detection. At the next step, a novel change detection algorithm based on the active contours, named desired feature local active contour (DFLAC), is presented. Finally, the accuracy and speed of the model are evaluated and compared with other studies.

## 2. Materials and Methods

In all change detection methods, at first, a difference image is produced from two images of the same region before and after the change using one of two standard operator subtraction or log-ratio. Next, a model is developed to detect the changed from the unchanged areas. In this paper, we first introduced a new difference image generation operator to increase the accuracy of the change detection method. On the other hand, we developed an innovative active contour model to extract changed regions from the difference image precisely.

Based on the active contours model proposed by Chunming Li [32], we introduced a new model that detects changes in the presence of inhomogeneity and noise in SAR images. We named the proposed model the desired feature local active contour (DFLAC) model because of the extraction of the desired target feature (changed regions) using the local image information. The DFLAC model needs training data from changed and non-changed areas, which are automatically produced by the Otsu thresholding method.

### 2.1. New Difference Image Operator

The first step of the change detection algorithm is to generate the difference image from two images before and after the change. Subtraction and log-ratio are two frequently used operators employed in many change detection research works [14]:

$$I_{d1} = |I_b - I_a|,$$ (1)

$$I_{d2} = \left| \log_{10}\left( \frac{I_b + 1'}{I_a + 1} \right) \right|,$$ (2)

where $I_d$, $I_b$, and $I_a$ are the difference, before and after images, respectively.

Although the subtraction operator can detect the small changes, the log-ratio has a better performance in the change detection of SAR images, which have multiplicative noises [14]. We proposed another equation for generating the difference image, namely the root multiplication log-ratio and normal difference (RMLND) operator. This operator fused two subtraction and log-ratio operators and has the benefits of them. Therefore, it can find out the small changes but has less sensitivity to SAR image noises (Figure 3):

$$I_{d3} = \sqrt{I_{d2} \times I_{d4}},$$ (3)

where $I_{d2}$ is log-ratio operator and $I_{d4}$ is a normal difference operator defined as follows:

$$I_{d4} = \left| \frac{I_b - I_a}{I_b + I_a + \eta} \right|,$$ (4)

where the parameter $\eta$ is a constant value that avoids wrong results in pixels, when $I_a$ and $I_b$ are equal to zero.

### 2.2. DFLAC Model

The proposed DFLAC model was developed based on Chunming Li's model [32], which detects changed regions from the difference image utilizing training data. In this model, the contour C of the model separates the difference image ($I_d$) domain (R) into changed and unchanged regions $R_c$ and $R_u$, so that $R = R_c \cup R_u$. The energy function of the model is the sum of the difference between the pixel values inside the curve C and training data. The function is minimized in an iteration process, and the curve C moves towards the border of the changed and non-changed regions. Inspired by Li's model, the energy function of the proposed model in a local region $S_x$ in the neighborhood of pixel x is defined as follows:

$$F(x) = \sum_{i=1}^{2} \int_A (I_d(y) - t_i)^2 dy, \tag{5}$$

where I(y) represents the difference image in $S_x$, and $t_i$ (i = 1,2) represents the training data of changed and unchanged regions. Additionally, A is defined as:

$$A = \begin{cases} S_x \cap R_c & i=1 \\ S_x \cap R_u & i=2 \end{cases}, \tag{6}$$

when more than one training data of changed and unchanged regions are introduced to the model. The minimum difference between image intensity and training data is used in the integral of Equation (5). Therefore, this equation can be changed as follows:

$$F(x) = \sum_{i=1}^{2} \int_A \min_j \left(I_d(y) - t_i^j\right)^2 dy, \tag{7}$$

where $t_i^j$ shows $j^{th}$ training data for the $i^{th}$ class of the image (changed and unchanged regions).

The above equation calculates the difference between the intensity of each pixel inside the curve C and the most similar pixel to it from the training data. Accordingly, after minimizing the energy function of the model, the curve C would extract changed regions from the unchanged area. Equation (7) is improved by utilizing a kernel function K such that K(x-y) = 0 for $x \notin S_x$ as a non-negative window function to separate $S_x$ from other image domains and make use of the local intensity in the energy function [32]:

$$F(x) = \sum_{i=1}^{2} \int_{S_x} K(x-y) \min_j \left(I_d(y) - t_i^j\right)^2 dy. \tag{8}$$

Due to the nature of the SAR images, the difference image $I_d$ is very heterogeneous and noisy. Therefore, we used the Li model to decompose the difference image into the true image, bias field (a parameter to formulate the intensity inhomogeneity in each pixel of the image), and noise:

$$I_d(y) = b(x).J(y) + n(y), \tag{9}$$

where $I_d(y)$ depicts the difference image, J(y) represents the true image, b represents the bias field, and n represents image noise [32].

In the $S_x$ area, the true image J(y), the training data consequently take approximately two constant values for changed and unchanged regions so that we can write:

$$t_i^j = b(x)p_i^j(y) + n(y), \tag{10}$$

where p represents the true value of training data. Therefore, based on Equation (10), Equation (8) will be presented as follows:

$$F(x) = \sum_{i=1}^{2} \int_{S_x} K(x-y) \min_j \left(I_d(y) - b(x)p_i^j(y) - n(y)\right)^2 dy. \tag{11}$$

The $F(x)$ is considered as the energy function of pixel x, and we should calculate the integral of $F_x$ to obtain the energy function of the whole image domain:

$$F = \int_\Omega \sum_{i=1}^{2} \int_{S_x} K(x-y) \min_j \left(I_d(y) - b(x)p_i^j(y) - n(y)\right)^2 dy dx. \tag{12}$$

After exchanging the order of the integration, Equation (12) is written as follows:

$$F = \int_{\Omega} \sum_{i=1}^{2} \int_{S_x} K(x-y)\min_j\left(I_d(y) - b(x)p_i^j(y) - n(y)\right)^2 dxdy. \tag{13}$$

The implicit form of curve C based on level set theory is used to minimize the energy function of the DFLAC model [33]:

$$F = \int_{\Omega} \sum_{i=1}^{2} \int_{S_x} K(x-y)\min_j\left(I_d(y) - b(x)p_i^j(y) - n(y)\right)^2 W_i(\varphi)dxdy. \tag{14}$$

In the above equation, the changed and unchanged regions can be presented by their membership functions defined by $W_1(\varphi) = H(\varphi)$ and $W_2(\varphi) = 1 - H(\varphi)$, respectively, where H represents the Heaviside function, and $\varphi$ represents a signed distance function to describe curve C implicitly [33].

The above energy function F, namely the image term of the DFLAC model, is used to regularize the model. The two additional terms called the length and distance regularization terms are added to the energy function introduced in the study by Li and his colleagues [32]:

$$F_{Final} = \alpha F_{image} + \beta F_{length} + \gamma F_{distance}, \tag{15}$$

where $\alpha$, $\beta$, and $\gamma$ represent the constant parameters that set the weight of each term in the energy function.

### 2.3. Minimization of the Energy Function

The parameter $\varphi$ is an implicit form of curve C, where $\varphi > 0$ depicts the inside of the curve and $\varphi < 0$ represents the outside of the curve and wherever $\varphi = 0$ shows the curve C of the model. As a result, changing the parameter $\varphi$ over time by minimizing $F_{Final}$ with respect to $\varphi$ using the standard gradient descent method, the position of the curve C is moved towards the border of the changed and unchanged regions of the image [34]:

$$\frac{\partial \varphi}{\partial t} = -\frac{\partial F_{Final}}{\partial \varphi}. \tag{16}$$

Therefore, the evolution equation of the level set function $\varphi$ over time is represented as follows:

$$\varphi_{k+1} = \varphi_k + \frac{\partial \varphi_k}{\partial t}\Delta t \Rightarrow \varphi_{k+1} = \varphi_k - \frac{\partial F_{Final}}{\partial \varphi}\Delta t, \tag{17}$$

where $\Delta t$ is the time step, and $\varphi_k$ represents a level set function in iteration k. The derivative of $\frac{\partial F_{Final}}{\partial \varphi}$ is calculated, and the corresponding gradient flow equation is presented as follows:

$$\frac{\partial \varphi}{\partial t} = -\alpha\delta(\varphi)\sum_{i=1}^{2} f_i + \beta\delta(\varphi)div\left(\frac{\nabla\varphi}{|\nabla\varphi|}\right) + \gamma div\left(d_p(|\nabla\varphi|)\nabla\varphi\right), \tag{18}$$

where $\delta(\phi) = \frac{\partial H(\phi)}{\partial \phi}$, and div is the divergence operator, $\nabla$ represents the gradient operator, and $d_p$ is defined as below according to the study by [32], as follows:

$$d_p(x) = \frac{p'(x)}{x} \quad \& \quad p(x) = \frac{(x-1)^2}{2}. \tag{19}$$

Additionally, the term $f_i$ can be achieved using the following expression:

$$f_i = \int K(x-y)\min_j\left(I_d(y) - b(x)p_i^j(y) - n(y)\right)^2 dx. \tag{20}$$

Moreover, the above integrals can be described using the convolution function so that:

$$f_i = \min_j \left( I_d{}^2 K_u - 2p_i^j I_d (b * K) - 2InK_u + \left(p_i^j\right)^2 \left(b^2 * K\right) + 2p_i^j n(b * K) + n^2 K_u \right), \tag{21}$$

where * represents the convolution operator and $K_u = \int K(y-x)dx$, where $K_u = 1$ except for the region near the boundary of the image domain $\Omega$ [32].

Furthermore, in order to calculate parameter b, we assume the parameters $\varphi$ and t are fixed parameters and minimize $F_{Final}$ with respect to b. So, the parameter b is given by:

$$b = \frac{\left((I_d - n)\sum_{i=1}^{2} P_i W_i(\varphi)\right) * K}{\left(\sum_{i=1}^{2} P_i^2\right) * K}, \tag{22}$$

where $P_1$ and $P_2$ are two matrixes, with the same size as the intensity matrix. Each element of these matrixes is a training data of changed and unchanged regions, respectively, that minimizes the E function for a given pixel:

$$E_i = (I_d - bt - n)^2 W_i(\varphi). \tag{23}$$

In the same way, other unknown parameters n and t are obtained as follows:

$$n = \frac{\left((I_d * K) - b\sum_{i=1}^{2}(P_i W_i(\varphi))\right)}{K_u}. \tag{24}$$

Furthermore, the true value of training data can be computed in each iteration as follows:

$$p_i^j = \frac{\int (b * K)(I_d - n)B_i^j W_i(\varphi)dx}{\int \left(b^2 * K\right)B_i^j W_i(\varphi)dx}, \tag{25}$$

where $B_i^j$ is a binary array, which indicates pixels that $p_i^j$ minimized the function $E_i$ defined in Equation (20). In step 1 of iteration, the training data $t_i^j$ used instead of $p_i^j$.

## 2.4. Training Data Sampling

The DFLAC model separates the difference image domain in two classes of changed and unchanged areas based on the training data of those classes. In the proposed model, the training data are not sampled from the difference image but are generated automatically. Firstly, a threshold number (T) is obtained from the difference image using the Otsu algorithm [35]. The number T is a normalized value that lies in the range [0, 1] that can be used to classify an image into two classes. Therefore, we use the Otsu' algorithm threshold (T) to generate the training data of the model from the difference image, automatically. Secondly, in the range of [T, 1], $k_1$ numbers and in the range [0, T], $k_2$ numbers with equal steps are selected as training data. It should be noted that $k_1$ and $k_2$ are the numbers of training data of the changed and unchanged classes, respectively. For example, if T = 0.6 and $k_1 = 2$ and $k_2 = 4$, then the numbers 0.8 and 1 are the training data of the changed class and the numbers 0, 0.15, 0.3, and 0.45 are selected as training data of the unchanged class. Thirdly, the produced training data can be projected into the other ranges, such as [0, 255]. It should be noted that each difference image operator has its specific threshold value (T) and training data.

### 2.5. Evaluation Indices

In order to evaluate the accuracy of the proposed model, three indices of the percentage correct classification (PCC), overall error (OE), and Kappa were used. PCC is the percentage of pixels that are correctly classified [36]:

$$PCC = \frac{TP + TN}{N}, \tag{26}$$

where TP and TN indicate the number of changed and unchanged pixels that are correctly classified, respectively. Additionally, N is the number of pixels in the image. OE is the sum of the number of changed and unchanged pixels incorrectly classified [36]:

$$OE = \frac{FP + FN}{N}, \tag{27}$$

where FP and FN are the numbers of unchanged and changed pixels, respectively, that are falsely detected. Finally, the Kappa coefficient is a parameter to indicate the accuracy of the classification model according to the difference between the observed accuracy and the chance agreement [36]:

$$Kapp = \frac{PCC - PRE}{1 - PRE}, \tag{28}$$

where:

$$PRE = \frac{(TP + FP).N_c + (TN + FN).N_u}{N^2}, \tag{29}$$

where $N_c$ and $N_u$ are the total numbers of pixels that belong to the changed and unchanged classes, respectively [36].

## 3. Implementation Results

### 3.1. Algorithm's Workflow

Figure 1 shows the flowchart of the proposed model. The proposed model has four main steps, including difference image generation, training data production, implementation of the DFLAC model, and finally, accuracy assessment. Furthermore, each step has several sub-steps that are described in the following.

- At the first step of difference image generation, two SAR images of the data set (before and after change) are introduced to the model, and the difference image is then produced based on one of Equations (1) to (4). Secondly, in the training data sampling step, a threshold T was first estimated using Otsu's method, then, the training data of changed and unchanged classes were selected based on this threshold. In the third step, the DFLAC model was implemented. The DFLAC model starts with defining the initial curve implicitly $(\varphi_0)$ based on a level set theory, which is a simple shape as a square and circle. Then, the evolution of the DFLAC model's curve was done over time using Equation (17). Next, the parameters b, n, and $p_i^j$ were then estimated according to Equations (22), (24), and (25). These last previous steps were repeated until the curve model reached stability and was not changed (i.e., $\int |\varphi_n - \varphi_{n-1}| < \varepsilon$). Finally, the output of the model was generated by separating changed regions (pixels inside the curve that $\varphi \geq 0$) from unchanged areas (pixels outside the curve that $\varphi < 0$).
- The accuracy assessment was the last step of the workflow, in which the error image was computed by subtracting the output image from the reference image as follows:

$$\text{error image} = |\text{Reference image} - \text{output image}|. \tag{30}$$

- Finally, the accuracy assessment of the model using the error map and some accuracy criteria, such as PCC, OE, and the Kappa, were estimated based on Equations (26)–(28).

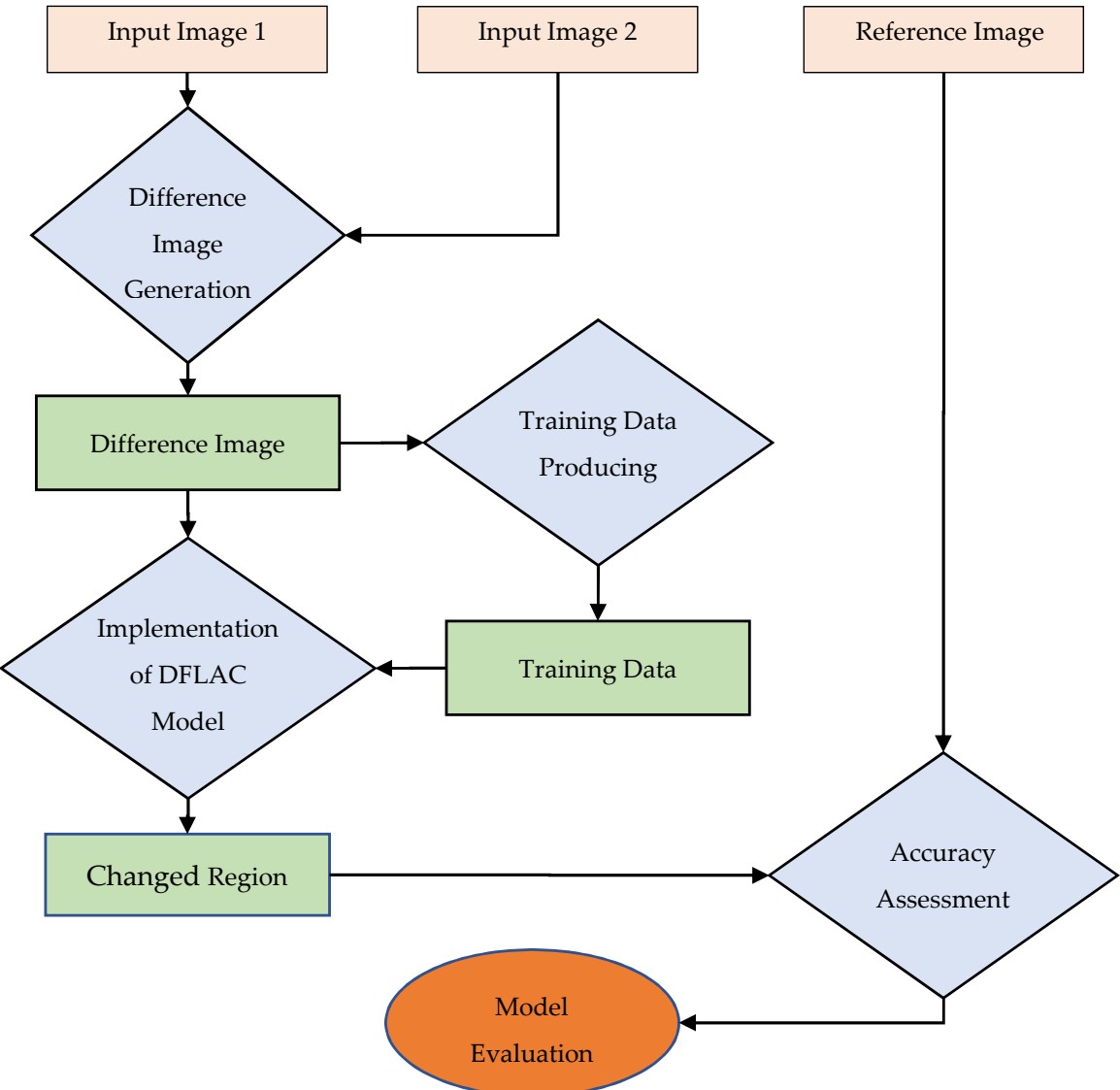

**Figure 1.** The flowchart of our proposed model.

The main stage in the proposed model is the implementation of the DFLAC model stage. Moreover, this step takes the most running time of the whole model. It is mainly because the evolution of the active contour is an iterative process that needs too much time to run. In this regard, more distinctions between the values of the changed and unchanged pixels in the difference image lead to a higher accuracy and speed of evolution of the active contour. In addition, the time step parameter, i.e., $\Delta t$ in Equation (17), regularizes the rate of the active contour evolution and has a considerable effect on the performance and speed of the model. Therefore, selecting a large value for the time step parameter leads to the passing of the model through the minimum of the energy function. Contrariwise, a very small value reduces the model speed. Accordingly, assigning an optimal value for the time step has a crucial impact on the efficiency of the model.

### 3.2. SAR Datasets

We used three sample data sets to evaluate the DFLAC model in change detection from SAR images. Brief information about the data sets is presented in Table 1.

**Table 1.** Characteristics of the SAR datasets.

| Data Set | Size (Pixel) | Resolution (m) | Date of the First Image | Date of the Second Image | Location | Sensor |
|---|---|---|---|---|---|---|
| Yellow River Estuary | 289 × 257 | 8 | June 2008 | June 2009 | Dongying, Shandong Province of China | Radarsat 2 |
| Bern | 301 × 301 | 30 | April 1999 | May 1999 | a region near the city of Bern | European Remote Sensing 2 satellite |
| Ottawa | 350 × 290 | 10 | July 1997 | August 1997 | Ottawa City | Radarsat 2 |

The first data set is a part of the Yellow River Estuary SAR data at Dongying, Shandong province, China. This data set shows a block of farmland that is landlocked. It should be noted that the first image of the Yellow River Estuary data set is four-look data, but the second image is single-look data, which means that the two images have different levels of speckle.

The second data set named Bern data was taken by the European Remote Sensing 2 satellite SAR sensor, which relates to a region near the city of Bern, Switzerland. River Aare flooded parts of the cities of Thun and Bern and the airport of Bern completely. Therefore, the Aare valley between Bern and Thun was chosen to extract flooded regions. The Ottawa data set is the third sample data set, which was acquired over the city of Ottawa by the Radarsat SAR sensor. This data set illustrates regions that were once flooded. Moreover, all data sets have a reference image as ground truth, which indicates changed regions precisely, and we used them to evaluate our model. Figure 2 illustrates the sample data sets and their reference image.

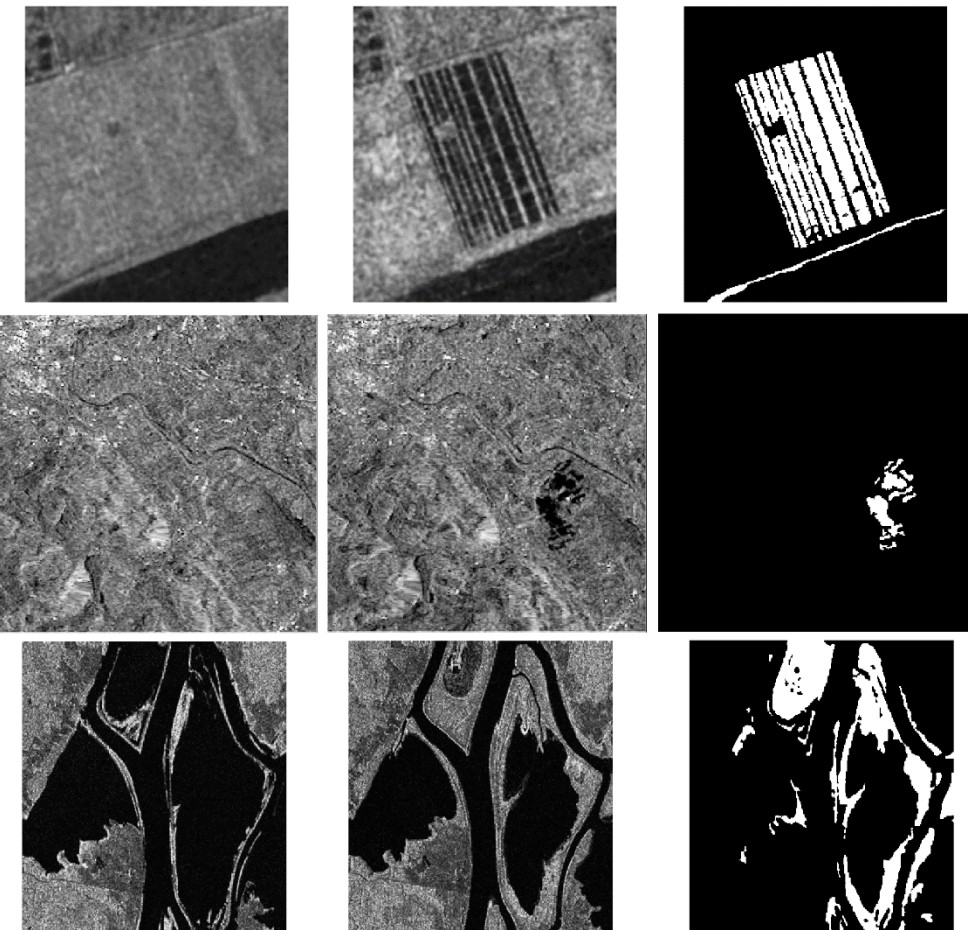

**Figure 2.** The SAR data sets include before change image, after change image, and reference change maps: (**Row 1**) Yellow River Estuary; (**Row 2**) Bern data set; (**Row 3**) Ottawa data set.

### 3.3. Difference Image

In this section, the difference images were generated using four formulas explained in Equations (1) and (2). The produced difference images lie on the range [0, 1], but for better processing, we normalized them in the range [0, 255]. Figure 3 demonstrates the difference image of the selected data sets.

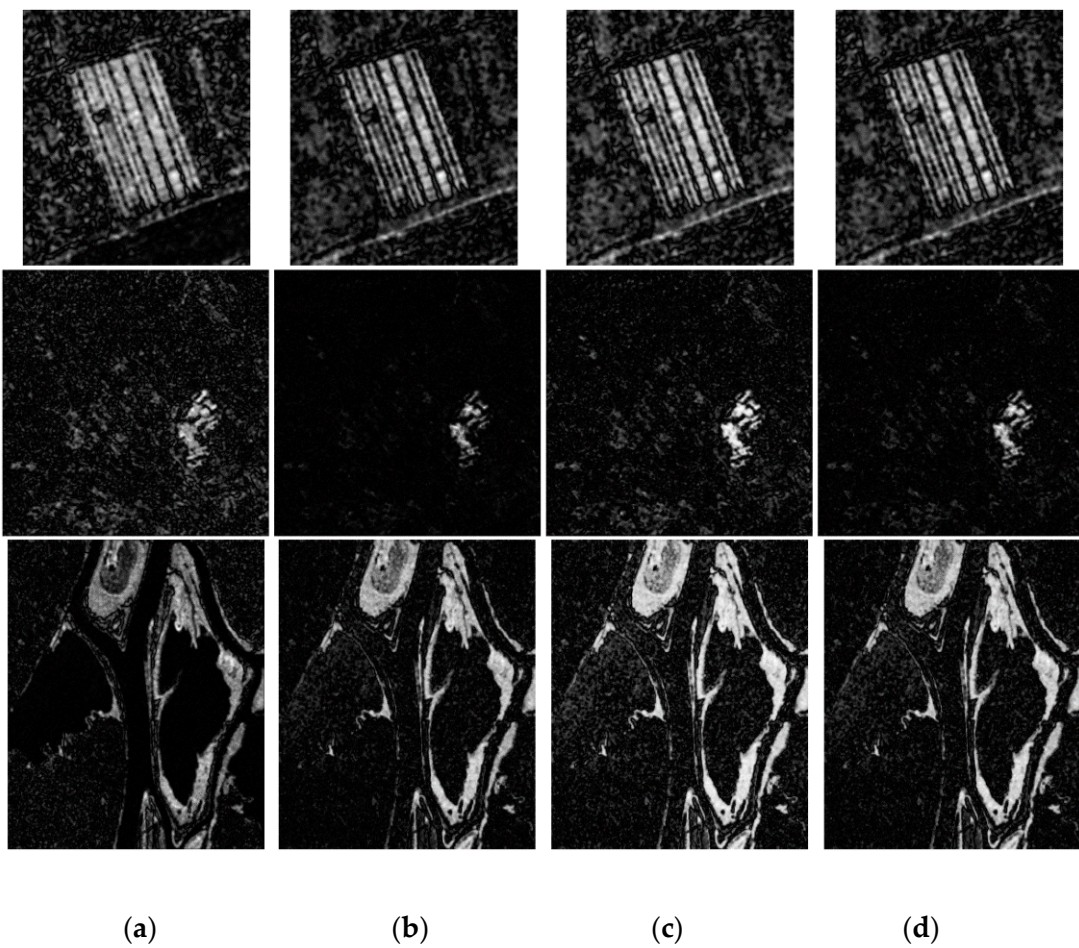

(**a**)　　　　　(**b**)　　　　　(**c**)　　　　　(**d**)

**Figure 3.** The difference images generated from four operators. (**a**) Subtraction; (**b**) Log-ratio; (**c**) normal difference; (**d**) RMLND.

### 3.4. Model Implementation

The constant parameters of the DFLAC model, i.e., $\alpha$, $\beta$, and $\gamma$, were determined by the trial and error method as 1, 0.11, and 0.4, respectively. In order to define the training data of changed and unchanged regions, we computed the Otsu thresholding algorithm on four difference image operators. Then, four numbers as training data of the changed class and two numbers as training data of the unchanged class were determined for each difference image operator in the range of 0 to 255 (Section 2.4). The numbers of training data of each dataset for the RMLND difference image operator are shown in Table 2.

**Table 2.** Training data of the sample data sets based on the Otsu threshold in the range [0, 255].

| Dataset | Changed Regions | | | | Unchanged Regions | |
|---|---|---|---|---|---|---|
| Yellow river Estuary | 127.00 | 169.67 | 212.33 | 255 | 0 | 42.167 |
| Bern | 112.40 | 159.94 | 207.47 | 255 | 0 | 32.44 |
| Ottawa | 127.00 | 169.67 | 212.33 | 255 | 0 | 42.17 |

Then, the difference image, constant parameters, and training data of each sample data set were introduced to the DFLAC model, and after 20 iterations, the curve of the model (C) extracted the changed regions. The final output of the DFLAC model based on the RMLND operator is represented in Figure 4.

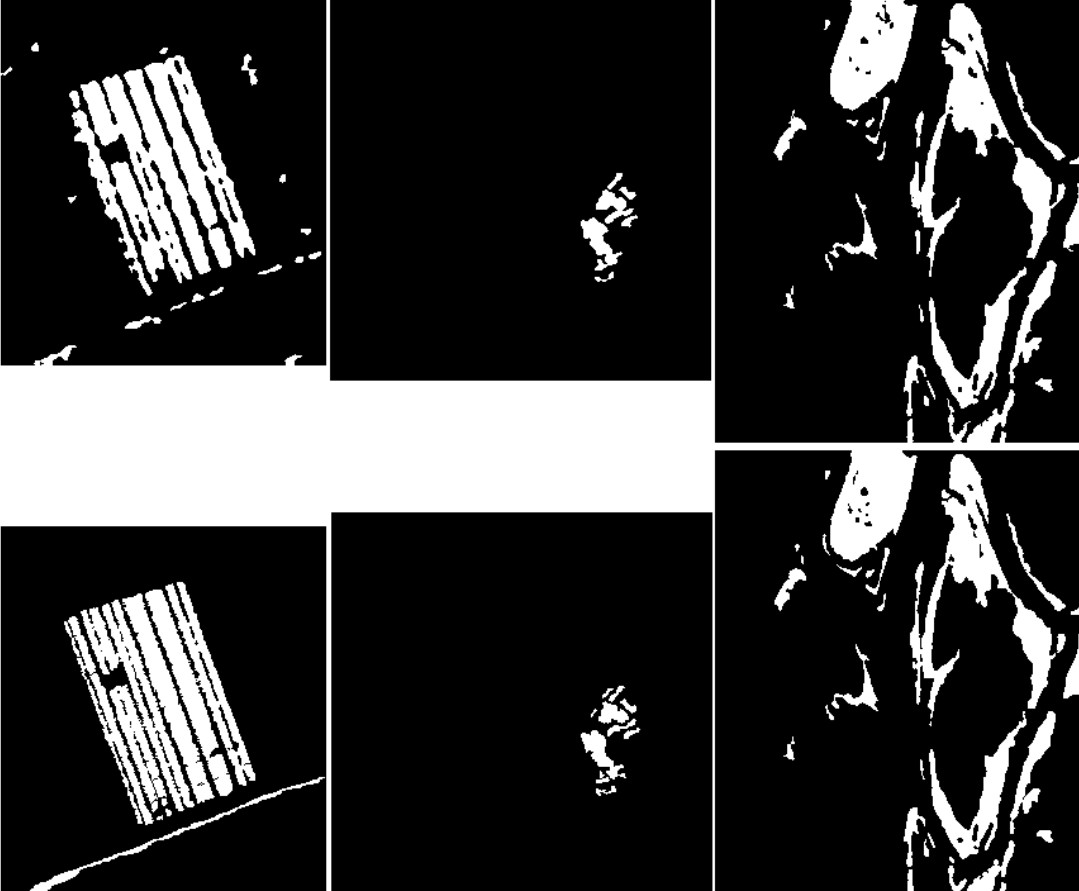

**Figure 4.** The results of the DFLAC model and the reference image (**Row 1**) Results; (**Row2**) Reference Images.

### 3.5. Accuracy of the Proposed Model

To evaluate the accuracy of the DFLAC model, we calculated the error image by subtracting the output of the model from the reference image. For this purpose, the accuracy parameters, such as Kappa, PCC, and OE, were calculated and compared with the results of the three most known models in change detection of SAR images, including saliency guided K-means (SGK) [14], neighborhood-based ratio (NR) [17], and log-normal generalized Kittler and Illingworth thresholding (LN-GKIT) [37]. Besides, two models of active contours, Chan and Vese (CV) [34] and distance regularized level set evolution (DRLSE) [38], were used for the evaluation of the proposed model. Table 3 demonstrates the accuracy parameters of the DFLAC model comparison with other algorithms for selected data sets by using the RMLND difference image operator as the best operator (Table 4).

As shown in Figure 4, due to the different levels of speckle in the two images of the Yellow River Estuary data sets, the difference between the two images in some areas of unchanged regions is relatively high. Therefore, the proposed model detects some unchanged area as a changed region and decrease the accuracy of the model.

**Table 3.** The accuracy of the proposed model compared to other models.

| Dataset | Method | PCC % | OE % | Kappa % |
|---|---|---|---|---|
| Yellow River Estuary | SGK | 98.06 | 1.92 | **85.24** |
| | NR | 88.33 | | 79.99 |
| | LN-GKIT | 69.60 | 30.82 | 33.78 |
| | CV | 95.37 | 4.63 | 84.38 |
| | DRLSE | 90.84 | 9.16 | 69.47 |
| | DFLAC | 95.49 | 4.51 | 84.65 |
| Bern | SGK | 99.68 | 0.32 | 87.05 |
| | NR | 99.66 | 0.34 | 85.90 |
| | LN-GKIT | 99.90 | 0.35 | 85.37 |
| | CV | 99.61 | 0.39 | 85.32 |
| | DRLSE | 98.70 | 1.30 | 63.32 |
| | DFLAC | 99.68 | 0.32 | **87.07** |
| Ottawa | SGK | 98.95 | 1.05 | 95.98 |
| | NR | 97.91 | 2.09 | 92.2 |
| | LN-GKIT | 98.35 | 2.22 | 91.87 |
| | CV | 97.06 | 2.93 | 88.92 |
| | DRLSE | 95.44 | 4.56 | 81.37 |
| | DFLAC | 99.00 | 1.00 | **96.26** |

**Table 4.** The comparison of the Kappa of the difference image operators.

| | Subtraction | Log-Ratio | Normal Difference | RMLND |
|---|---|---|---|---|
| Yellow River Estuary | 75.22 | 83.97 | **84.71** | 84.65 |
| Bern | 32.44 | 81.26 | 83.14 | **87.07** |
| Ottawa | 83.81 | 95.26 | 94.92 | **96.26** |
| Average | 63.82 | 86.83 | 87.59 | **89.33** |

## 4. Discussion

In this section, we discuss the parameters that affect the accuracy of the model. Additionally, to evaluate the efficiency of our model, the speed of the model is compared with the SGK approach [14].

### 4.1. Accuracy Assessment

The accuracy and performance of the proposed model depend on three parameters, including the fixed parameters, i.e., $\alpha$, $\beta$, and $\gamma$, in Equation (18), the difference image operator type, and the number of training data in which the impact of each is discussed below.

#### 4.1.1. The Constant parameters

The constant parameters $\alpha$, $\beta$, and $\gamma$ regularize the effect of the length, distance, and image terms in the energy function of the model (Equation (12)). Changing these parameters causes a change in the evolution of the curve and the results of the model. As a result, the accuracy of the model is affected. To determine the role of these parameters in the accuracy of the model, we fixed two parameters and then changed the other parameters with 0.1 steps. We then calculated the average accuracy of the three data sets. The rate of change of the Kappa parameter due to the variation of the constant parameters $\alpha$, $\beta$, and $\gamma$ is depicted in Figures 5–7, respectively.

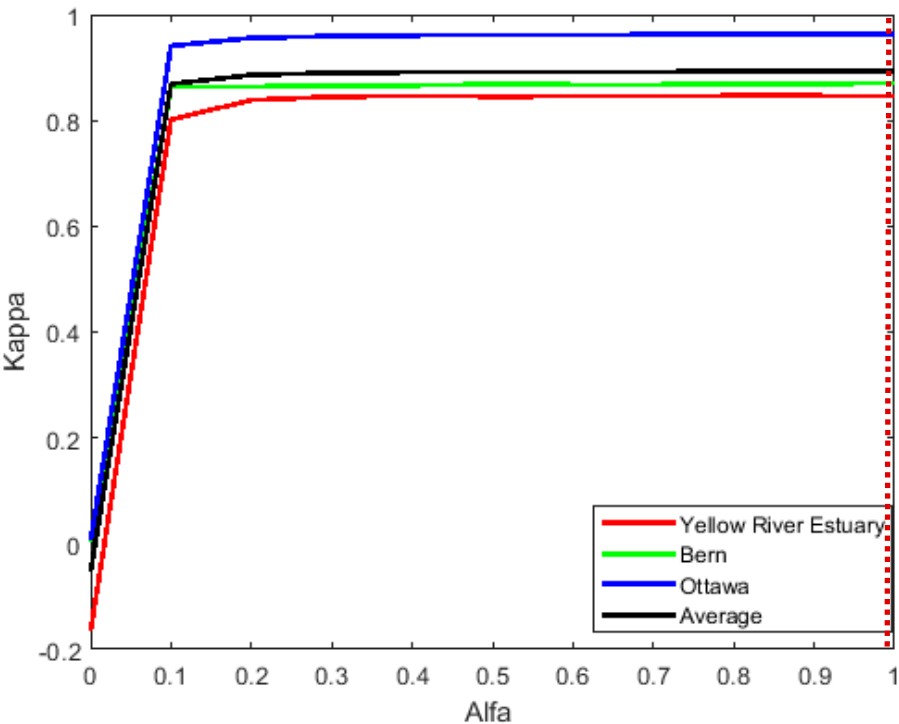

**Figure 5.** The variation of the Kappa according to the $\alpha$ parameter.

Based on Figure 5, in all data sets, the Kappa increases when $\alpha$ rises from zero to 1. Therefore, 1 is the best value for parameter $\alpha$. Additionally, as shown in Figure 6, the change rate of Kappa with respect to $\beta$ in all data sets is low because the impact of the length term in the energy function of the active contour models is slight. Figure 7 shows that the Yellow River Estuary data set has its maximum Kappa $\gamma = 0.2$ contrast with other data sets and the average of all data sets, which reached its peak in 0.4.

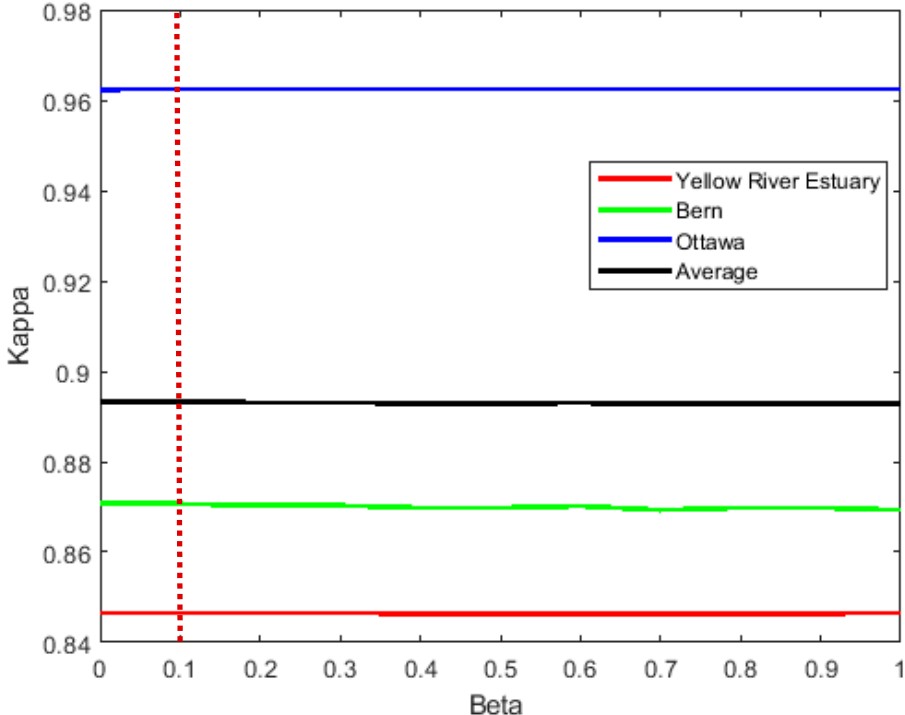

**Figure 6.** The variation of the Kappa according to the $\beta$ parameter.

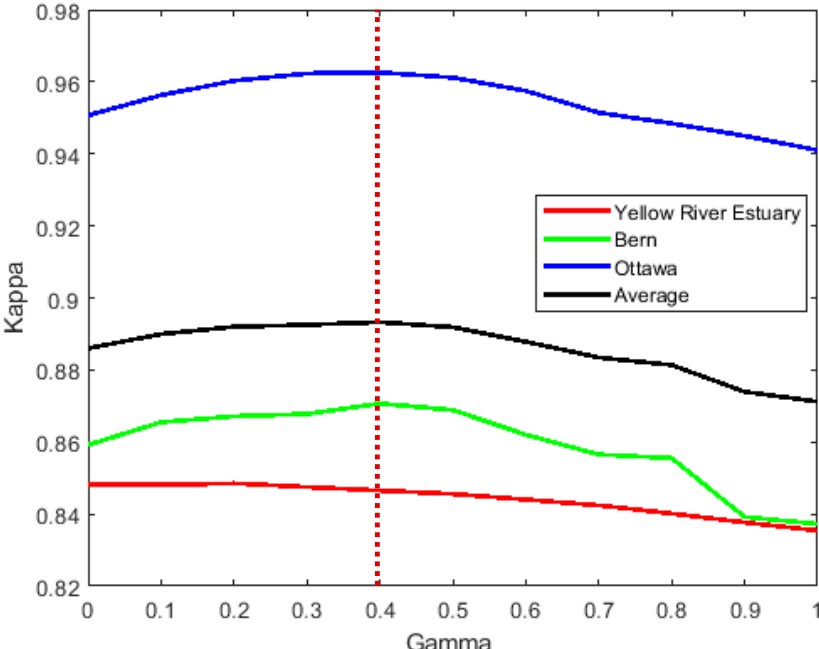

**Figure 7.** The variation of the Kappa according to the γ parameter.

Consequently, the maximum value of Kappa occurs when α, β, and γ are 1, 0.11, and 0.4, and therefore, we selected these numbers as optimal values of the constant parameters for implementing the model. It can be noted that the Yellow River Estuary and Bern data sets have minimum and maximum sensitivity to the variation of the constant parameters, respectively.

### 4.1.2. The Difference Image Operator Type

We assessed the four difference-image operators, including subtraction, log-ratio, normal difference, and RMLND, in the case of the Kappa parameter. The results and statistics of these operators, which were executed on sample data sets, are illustrated in Table 4 and Figure 8.

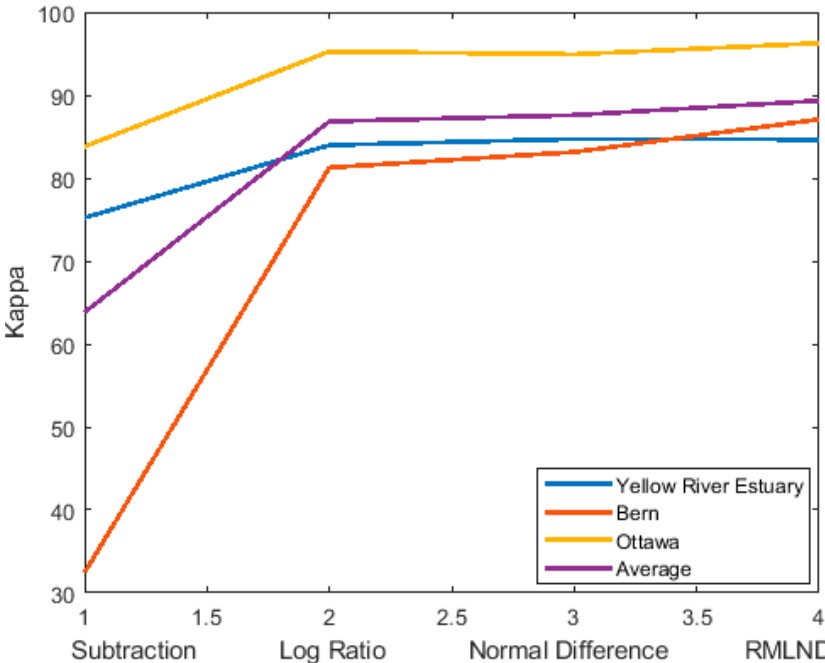

**Figure 8.** The variation of the Kappa for all operators in each data set.

It can be seen that the RMLND operator has the best performance compared to the other operators. Therefore, the RMLND difference image was chosen as the principal operator for implementing our proposed model. Additionally, the normal difference model, after the RMLND operator, has the best efficiency; the log-ratio and the subtraction models are in the next ranks, respectively. The reason for the low performance of the subtraction model is that it detects small changes due to speckle noise. This defect is very significant at the border of the changed regions.

According to Figure 8, the normal difference operator achieved the best result for the Yellow River Estuary data set, and after that, the RMLND, log-ratio, and subtraction operators have more accuracy, respectively. Moreover, the subtraction operator obtains the worst results for the Bern data set 32.44%) and the accuracy of the log-ratio, normal difference, and RMLND operators increases, respectively. Additionally, the RMLND operator has the best results for the Ottawa image, and the accuracy of the log-ratio, normal difference, and subtraction is in the next ranks. In addition, according to the mean accuracy of the three data sets, the best operator is RMLND, and after that, the normal difference and log-ratio have a better performance, respectively. Finally, the subtraction operator achieved less accuracy compared to the rest of the operators.

### 4.1.3. Numbers of Training Data

The numbers of training data of changed and unchanged regions is an effective parameter in the accuracy of the proposed model. In order to evaluate the effect of the numbers of training data, and determine the optimal numbers of the data, we fixed the numbers of unchanged data and calculated the mean of the Kappa of data sets relative to several training data of changed areas. Similarly, the number 2 was obtained as the best number of the training data in the unchanged regions. Figures 9 and 10 demonstrate the variation of Kappa with respect to the numbers of training data of the changed and unchanged areas.

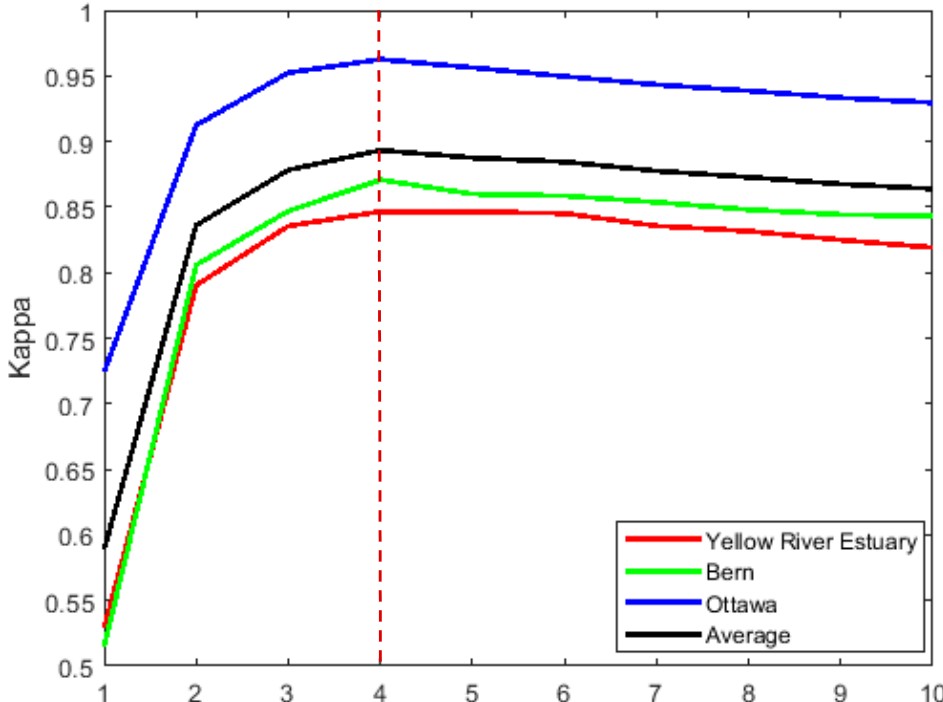

**Figure 9.** The Kappa change rate with respect to the number of training data of the changed region.

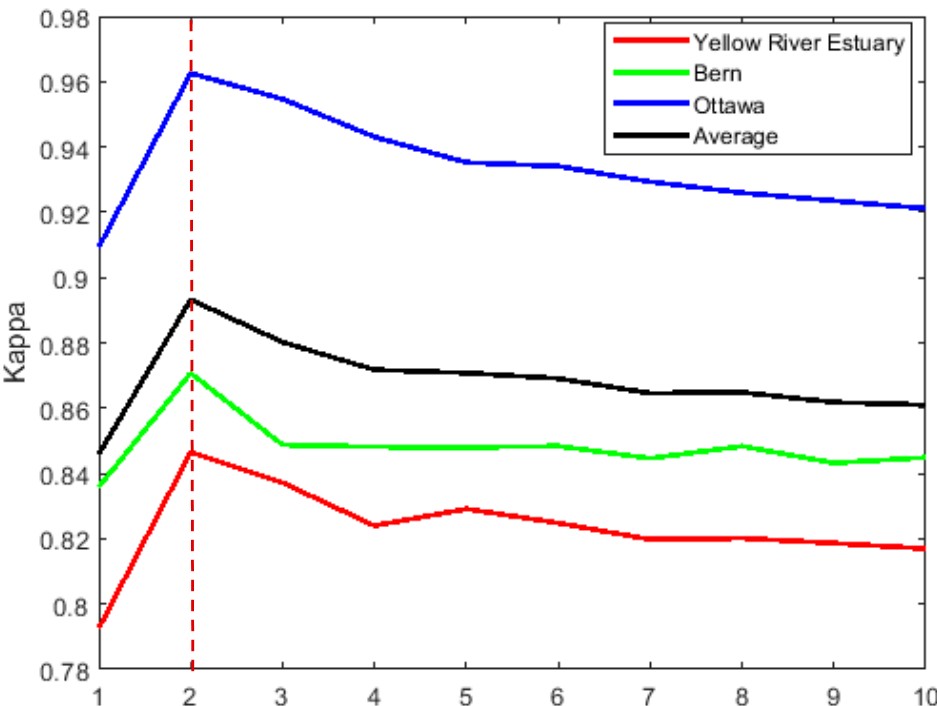

**Figure 10.** The Kappa change rate with respect to the number of training data of the unchanged region.

Based on Figure 9, the kappa parameter increases with an increasing number of classes of the changed regions in all data sets and reaches its maximum value in four, and then decreases gradually. According to Figure 10, the Kappa of all data sets and the average rise sharply by increasing the number of training data of the unchanged regions from 1 to 2, then it goes down slightly. Therefore, the maximum Kappa is related to the optimal numbers of training data of the changed and unchanged region, which the four and two number is the best, respectively.

### 4.2. Running Time Comparison

One of the efficiency parameters of a model is its running time. Since the proposed model has five steps, the running time of each step was estimated separately (Table 5). Therefore, we compared the run time of the proposed model and the SGK model [14] in three sample data sets. Table 5 illustrates the run time of two models in each data set. As seen in Table 5, our model is approximately 10 times faster than the SGK model in the three sample data.

**Table 5.** The comparison of the model's speed in the second.

| Data Sets | DFLAC Time Steps | | | | | | SGK | SGK/DFLAC |
|---|---|---|---|---|---|---|---|---|
| | 1 | 2 | 3 | 4 | 5 | Total Time | | |
| Yellow River Estuary | 0.01 | 0.04 | 0.01 | 8.90 | 0.01 | 8.97 | 60.53 | 6.75 |
| Bern | 0.02 | 0.05 | 0.01 | 8.31 | 0.01 | 8.40 | 86.60 | 10.31 |
| Ottawa | 0.02 | 0.05 | 0.02 | 9.34 | 0.01 | 9.44 | 101.17 | 10.72 |
| Average | 0.02 | 0.05 | 0.01 | 8.85 | 0.01 | 8.94 | 82.77 | **9.25** |

### 5. Conclusions

In this paper, we proposed a novel model of active contours for change detection of SAR images. The model was designed to extract a target feature in a digital image by getting some training data from the target feature and image background. Therefore, in this paper, the changed and unchanged features were considered as target features and image backgrounds, respectively. Besides, the training data were generated from the difference image using the Otsu thresholding method automatically.

Furthermore, we introduced a new difference image operator to attain more accuracy compared to the existing operators. For the accuracy assessment of the model, it was applied to three temporal SAR images, and the outputs were compared to their corresponding reference images (ground truth). The accuracy of the proposed model depends on three constant values of the model, which were determined by the trial and error method. Additionally, the number of training data of the changed and unchanged region, which were identified manually, affect the accuracy of the model. The results of the model demonstrate the higher accuracy of the proposed model compared with the five most known models of change detection in SAR images. It should be mentioned that the proposed model was completely implemented in the Matlab R15b software environment.

**Author Contributions:** Conceptualization, methodology, and implementation of the programing: S.A.; Original draft preparation: S.H. and S.A.; Result evaluation: S.H.; Review and editing: S.H. All authors have read and agreed to the published version of the manuscript.

**Funding:** This research received no external funding.

**Acknowledgments:** The authors would like to thank Kamran Kazemi, associate professor of the Electrical Engineering Department, Shiraz University, Shiraz, Iran, for providing the data sets used in this study.

**Conflicts of Interest:** The authors declare no conflict of interest.

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
