# Peer review of "A Novel Active Contours Model for Environmental Change Detection from Multitemporal Synthetic Aperture Radar Images"

_remotesensing, doi:10.3390/rs12111746_

Round 1
Reviewer 1 Report
1. From Fig.1, it can be seen that, there are some steps. In Table 4, does the time include all the steps, or just the DFLAC model? If not the whole time, it is unfair for other methods. So if possible, please list the time consumption of each step. 2. Actually, there are many ACM algorithm and its variants, and there are also many change detection methods, but in the comparison experiment, only one method is compared, which is not enough. At least, 2 ACM methods and two other methods for change detection. 3. For the training data, I still have a question. There are only three images which are not very large. So if the training data are from these 3 images, does it mean that the training data and the test data are both from one same data? 4. Please describe the training with more detail and show some training data samples. 5. In fig.9 and 10, what is the mean of the abscissa? 1 means only one training data?Author Response
Please see the attached file.

Reviewer 2 Report
In this paper it is presented a novel approach for image change detection with supervised training.
My first recommendation is to better explain what represents for you training data, it is not clear through the methodology explanation of paragraph 2.2
At the beginning of paragraph 2.2 I supposed they are images with labeled changed (1) and unchanged (0) areas, but in this case it is not clear why you compare it with I instead of Id (which is between 0 and 1).
In line 129 you define I as the "intensity of difference image" in Sx, in line 145 I is defined as the "observed image". Please clarify
In eq 10, it looks like the training data are not picked directly from some images but they are modeled somehow, but in paragraph 2.4 the authors say the training data are picked from some classified images. Please clarify
The authors compare their approach with the SGK method, I would suggest to better explain why this algorithm has been selected for the comparison and a short description of it.
Minor considerations:
typo pag 3, line 109 "tow"
pag 3, line 129, not clear what I represents, also because you redefine what I represents in line 145
pag 3, eq 5, what represents A in the equation? I don't see the dependency on the contour C
pag4, line 141-148, sentence not clear
pag 9, line 247, I would anticipate in this section the methodology used to estimate the parameters alpha, beta, gamma (4.1.1), otherwise it is hard to imagine how you could select these 3 values with trial and errors.
Reviewer 3 Report
Dear authors, based on the corrections in the attached document I propose medium revision for the MS, becasue I believe your work is good and if you present it in more details it will have greater impact in the scientific community.
Kind regards

Reviewer 4 Report
I have had the pleasure to read your manuscript and it is certainly an interesting piece of work, which has been well-prepared and documented wrt. methodology. This is "as far as I can see", I should add, as I would require more time to investigate the equation development. I am thus not referring to this core part, and hope other reviewers did so.
Apart from minor spelling mistakes or issues with the sentence structure (L104 sentence, L109 tow, L141 sentence, referring equation in L153, first sentence in the conclusions), I do have some issues with respect to the presentation of results.
Results
I don't think it makes so much sense to itemize the workflow steps and present the same contents graphically without much added value. The graphical workflow is certainly needed, but I would appreciate a decent description of the workflow along with the description of some experiences and perhaps comments on processing time. The text description should then refer to workflow steps depicted in the diagram.
Test datasets:
Is there any reason why you haven't used quite popular Sentinel data? Did I miss a comment here?
Data Contents and Accuracy Assessment
Could you please introduce the image scenes by explaining what we can see, along with proper geographical references and scale information? Am I wrong when I assume that the estuary image shows either a solar-pane array or some aqua-culture basins?
At least Bern and Ottawa show water surfaces? Does that mean that your algorithm was able to separate water bodies from non-water bodies in SAR data? How much sense does that make to demonstrate the feasibility of a change-detection algorithm? How about adding deforestation, desertification or other environmental effects? Right now it is a water-body change detection algorithm? Would you disagree?
Based on the selection of test datasets I am not convinced that the accuracy assessment is very representative. I do not doubt your approach but I believe it is not at all well presented here.
Finally, the conclusions lack any information about available code, implementation environment and so on. I am a strong believer that a modern paper should address this matter carefully.
Round 2
Reviewer 1 Report
All my concerns have been solved, so this manuscript can be accepted.
Author Response
Dear Reviewer;
We appreciate your constructive comments and helpful feedback.
Reviewer 2 Report
The authors made the corrections required by the reviewers, this new version has been significantly improved.
Author Response

(The authors gave the same response as above.)

Reviewer 4 Report
Dear authors,
thank you for addressing my comments.
I still do have two concerns.
I understand that the three scenes are used in use-cases and benchmark-testing in the literature. However, if that is the case someone or some institute must have made them available as you clearly did not download them yourself. There must be a reference in how far these scenes are qualified to be some sort of processing "standard" and you will understand that I continue to raise this concern as the scene selection is quite limited and not representative of the real-life environmental changes that we face (and which is the title of your manuscript).
Secondly, you have not addressed my concerns regarding code availability, and I can imagine why. Now, see this review process from the perspective of the reviewer:
- I receive a manuscript discussing the details of an algorithms and its performance.
- the images used are lacking any verification, not even the location is properly known. You claim that as long there is change, the algorithm will detect it. How do we know images are not altered in any way?
- There is no code.
Now, how could anyone verify anything that you write, except for the development of equation and procedures that you elaborate on (in a one-week time frame, I should add)?
So, I would recommend addressing this matter in any constructive way, even if you do not plan to release any code.
Kind regards.
